

# Association analysis of MTHFR (rs1801133 and rs1801131) gene polymorphism towards the development of type 2 diabetes mellitus in Dali area population from Yunnan Province, China

Yongxin Liu[1,*], Genyuan Pu[1,*], Caiting Yang[1], Yuqing Wang[1], Kaitai Jin[1], Shengrong Wang[1], Xiao Liang[1], Shenghe Hu[2], Shuguang Sun[2] and Mingming Lai[1]

[1] School of Basic Medicine, Dali University, Dali, Yunnan, China, Dali, China
[2] The First Affiliated Hospital of Dali University, Dali, Yunnan, China, Dali, China
[*] These authors contributed equally to this work.

## ABSTRACT

**Background**. Type 2 diabetes mellitus (T2DM) is a common complex metabolic disorder that exhibits a strong genetic predisposition. 5,10-methylenetetrahydrofolate reductase (MTHFR) regulates folate metabolism, which has been proposed to be associated with T2DM, although the relationship is inconsistent among different geographical areas. This study aimed to investigate the effects of MTHFR C677T (rs1801133) and A1298C (rs1801131) loci polymorphisms on T2DM susceptibility in the population of the Dali area in Yunnan Province, China.

**Methods**. This case-control study included 445 patients with T2DM and 272 healthy control individuals from the Dali area of Yunnan Province. Genotyping of the MTHFR gene polymorphisms was performed using the competitive allele-specific PCR (KASP) method. The effects of genetic variations of the MTHFR gene on T2DM risk were evaluated using odds ratios (OR) and 95% confidence intervals.

**Results**. The results of the present study revealed that the TT genotype (OR = 1.750, $P = 0.030$) and the T allele (OR = 1.252, $P = 0.047$) at the MTHFR C677T locus were considerably associated with the increased odds of developing T2DM. In addition, the CC genotype (OR = 3.132, $P = 0.032$) at the MTHFR A1298C locus also substantially increased the odds of developing T2DM. The T-A haplotype (OR = 1.305, $P = 0.030$) of MTHFR C677T and A1298C exhibited the increased odds of developing T2DM. Biochemical index analyses showed that patients with T2DM who carried the CT or TT genotype of MTHFR C677T expressed substantially higher levels of fasting blood glucose (FBG), homocysteine (Hcy), and tumor necrosis factor-alpha (TNF-$\alpha$) than those of the CC genotype. Moreover, the FBG and Hcy levels were considerably higher in patients with T2DM who carried the CC or AC genotype of MTHFR A1298C than those of the AA genotype. No obvious association was observed between these MTHFR polymorphisms and cardiovascular risk in T2DM.

Corresponding authors
Shuguang Sun, sshuglily@163.com
Mingming Lai, jz1507@dali.edu.cn

**Conclusion**. Our study suggests that the genetic variations of MTHFR C677T and A1298C are significantly associated with T2DM susceptibility in the population of the Dali area of Yunnan Province, China.

## INTRODUCTION

Type 2 diabetes mellitus (T2DM) is one of the most prevalent metabolic diseases and is characterized by high glucose levels and insulin resistance. In China, approximately 140 million adults suffer from diabetes mellitus, and this number will reach 170 million by 2045 (*Magliano, Boyko & IDF Diabetes Atlas 10th edition scientific committee, 2021*). It has been verified that the progression of T2DM is remarkably affected by multiple genetic factors. Genetics can determine the susceptibility of an individual to developing T2DM (*Tinajero & Malik, 2021*). Therefore, the identification of inheritable susceptibility genes for T2DM can provide valuable information and a reference for the clinical prevention of T2DM.

Evidence has indicated that individuals with insufficient intake of folic acid, also known as folate, vitamin B9, and vitamin B11, are more likely to suffer from T2DM (*Hayden & Tyagi, 2021*). 5,10-methylenetetrahydrofolate reductase (MTHFR) is a key enzyme involved in folate metabolism. MTHFR, located at human chromosome 1p36.3, participates in the catalysis of 5,10-methylenetetrahydrofolate to 5-methyltetrahydrofolate (*Goyette et al., 1994*). The decrease of this enzyme activity leads to the increase of homocysteine (Hcy) level, which induces vascular endothelial injury and dysfunction and contributes to cardiovascular diseases, diabetes, tumors, and other diseases (*Ponce-Ruiz et al., 2020*; *Lapik, Ranjit & Galchenko, 2021*; *Gupta et al., 2018*; *Wang et al., 2022*; *Fan et al., 2020*; *Nasri et al., 2019*). The MTHFR gene has numerous single nucleotide polymorphisms (SNPs) while two of them, rs1801133 (MTHFR C677T) and rs1801131 (MTHFR A1298C) were extensively investigated concerning the risk of T2DM, although the results were conflicting (*Pathak et al., 2022*; *Shorikov et al., 2022*; *El Alami et al., 2024*). MTHFR C677T polymorphism showed a more significant association with T2DM in the Asian population than in the European and African populations (*Meng et al., 2019*). Such differences might be caused by different genetics and environment. Therefore, the association between these variations and T2DM needs to be verified in different areas and ethnic groups.

The excessive oxidative stress produced by abnormalities in glucose metabolism usually found in T2DM patients (*Darenskaya, Kolesnikova & Kolesnikov, 2021*). It was reported that single nucleotide polymorphisms (SNPs) of genes encoding the antioxidant enzymes, such asglutathione peroxidase (GPx), catalase (CAT), superoxide dismutase (SOD), and paraoxonase (PON), affect the oxidative stress (*Oh et al., 2007*; *Min et al., 2006*; *Senoner & Dichtl, 2019*; *Abd El Azeem et al., 2021*; *Wang et al., 2014*). Previous studies have demonstrated thatfolate was involved in metabolic reactions for combating oxidative stress (*Asbaghi et al., 2021*; *Jakovljevic Uzelac et al., 2020*; *Zhou et al., 2024*). However, none of
these studies investigated the effect of MTHFR gene polymorphism on oxidative stress in patients with T2DM.

The Dali Bai Autonomous Prefecture is located in the northwest of Yunnan Province, China. So far, few studies have investigated the genetic susceptibility to developing T2DM in the population of the Dali area. Here, we conducted a case-control study to reveal the potential association of MTHFR C677T (rs1801133) and A1298C (rs1801131) loci polymorphisms with T2DM in the population of the Dali area of Yunnan Province, China.

## MATERIALS AND METHODS

### Study subjects

A total of 445 patients with T2DM and 272 unrelated healthy control individuals were recruited from the First Affiliated Hospital of Dali University from January 2021 to July 2023. Patients with T2DM were diagnosed according to the Chinese Medical Association Criteria with fasting plasma glucose of ≥7.0 mmol/L (*Expert Committee on the Diagnosis and Classification of Diabetes Mellitus, 2003*). Patients suffering from any of the following conditions were excluded from the study: type 1 diabetes mellitus, gestational diabetes mellitus, special type diabetes mellitus, malignant tumor, autoimmune disease and mental illness. The control subjects were screened from a physical examination, who were gender-, age-, and region- matched with fasting plasma glucose of 3.9–6.1 mmol/L. The exclusion criteria for the control subjects were diabetes mellitus, hypertension, hyperlipidemia, hepatic and renal diseases, autoimmune diseases. Patients with T2DM and CVD were diagnosed according to guidelines for the management of cardiovascular disease in patients with diabetes, which included coronary heart disease, arrhythmia, valvular heart disease, hypertension, and hypertension heart disease (*Marx et al., 2023*). All included individuals had 1–2 mL of whole blood collected, and their liver, kidney function, and glycolipid metabolism parameters were evaluated. The assessment of liver and kidney function parameters included total protein (TP), albumin (ALB), globulin (GLB), albumin/globulin (ALB/GLB), blood urea nitrogen (BUN), serum creatinine (Scr), and serum uric acid (SUA). The tested glycolipid metabolism parameters included fasting blood glucose (FBG), total cholesterol (TC), triglyceride (TG), high-density lipoprotein cholesterol (HDL-C), low-density lipoprotein cholesterol (LDL-C), apolipoprotein A1 (APOA1), and apolipoprotein B (APOB). This study was reviewed and approved by the Medical Ethics Committee of Dali University (No. 202010-21). All participants provided written informed consent to participate in this study.

### Genotyping assay

Genomic DNA was extracted from 200 µL of whole blood from each individual using a genomic DNA kit (Tiangen Biotech Co., Ltd., Beijing, China, Lot Number: DP304) according to the manufacturer's instructions. DNA was quantified using a NanoDrop One (Thermo Scientific Corporation, Waltham, MA, USA) and diluted to 20–50 ng/µL. Genotyping of MTHFR C677T (rs1801133) and MTHFR A1298C (rs1801131)

polymorphisms was performed using the method described in our previous study (*Liu et al., 2023*).

### Detection of serum Hcy, TNF-α, IL-6, SOD, and MDA levels

The levels of Hcy, tumor necrosis factor-alpha (TNF-α), and interleukin 6 (IL-6) in the serum were detected using commercial ELISA kits (Hcy: Wuhan Huamei Bioengineering Co., Ltd., Wuhan, China, Lot Number: CSB-E08895 h; TNF-α: NOVUS., Lot Number: VAL105; IL-6: NOVUS., Lot Number: VAL102) according to the manufacturer's instructions. The level of superoxide dismutase (SOD) in the serum was detected using the nitrogen blue tetrazolium colorimetric method (Beyotime Biotechnology Co., Ltd., Jiangsu, China, Lot Number: S0109). The concentration of malondialdehyde (MDA) in the serum was assessed using the thiobarbituric acid method (Beyotime Biotechnology Co., Ltd., Jiangsu, China, Lot Number: S0131S).

### Statistical analysis

We used SPSS v.25.0 software (SPSS, Inc., Armonk, NY, USA) for all statistical analyses. Variables were assessed for normality using the Kolmogorov–Smirnov Z test. The Student's $t$-test, Wilcoxon signed rank test, and $\chi^2$ tests were used to evaluate the case-control differences in the distribution of risk factors. The Hardy-Weinberg equilibrium was used to determine whether the selected study participants were representative of the population. The GPower 3.1.9.7 software was used to calculate the power of the sample size. Linkage disequilibrium and haplotype analysis were performed using the SHEsisPlus online software (http://shesisplus.bio-x.cn/SHEsis.html). A two-tailed $P < 0.05$ was considered statistically significant.

## RESULTS

### General clinical characteristics of the study subjects

As shown in Table 1, this study consisted of 445 patients with T2DM (262 males and 183 females) with an average age of 56.06 ± 12.10 years (range: 24–94) and 272 unrelated healthy control individuals (141 males and 131 females) with a mean age of 54.19 ± 13.50 years (range: 28–88). There were no significant differences between the T2DM and control groups with regard to age or gender. However, the BMI of T2DM group was significantly higher than that of control group. Other biochemical variables, such as TP, ALB, GLB, ALB/GLB, BUN, FBG, TG, HDL-C, and APOA1, differed between these two groups. As expected, the T2DM group presented a higher prevalence of hyperlipidemia compared to the control group.

### Association of MTHFR C677T and A1298C polymorphisms with the risk of developing T2DM

The competitive allele-specific PCR (KASP) method was used to evaluate the MTHFR C677T and A1298C polymorphisms (Fig. 1). The Hardy-Weinberg equilibrium was used to assess the control group, and no significant deviation was identified (Table S1). Furthermore, the power of the sample size in this study was greater than 0.9 in both the T2DM and control groups. As shown in Table 2, both MTHFR C677T and

**Table 1** Comparison of clinical variables between the T2DM and control groups.

| Variables | Control ($n = 272$) | T2DM ($n = 445$) | *P*-value | *P'*-value |
|---|---|---|---|---|
| Age (years) | 54.19 ± 13.50 | 56.06 ± 12.10 | 0.063 | 1 |
| Gender | | | | |
| Male, n (%) | 141 (51.84) | 262 (58.88) | – | |
| Female, n (%) | 131 (48.16) | 183 (41.12) | 0.065 | 1 |
| BMI | 23.92 ± 3.29 | 24.83 ± 3.52 | 0.001** | 0.018* |
| Hyperlipidemia, n (%) | 80 (29.41) | 248 (55.73) | <0.001*** | 0.018* |
| TP (g/L) | 75.44 ± 4.49 | 68.25 ± 7.16 | <0.001*** | 0.018* |
| ALB (g/L) | 45.63 ± 2.64 | 40.39 ± 5.61 | <0.001*** | 0.018* |
| GLB (g/L) | 29.80 ± 3.85 | 27.84 ± 4.83 | <0.001*** | 0.018* |
| ALB/GLB | 1.50 ± 0.30 | 1.50 ± 0.40 | 0.022* | 0.396 |
| BUN (mmol/L) | 5.29 ± 1.32 | 5.81 ± 2.54 | <0.001*** | 0.018* |
| Scr ($\mu$mol/L) | 70.00 ± 26.00 | 69.00 ± 26.00 | 0.124 | 1 |
| SUA ($\mu$mol/L) | 348.33 ± 95.43 | 341.28 ± 97.79 | 0.349 | 1 |
| FBG (mmol/L) | 4.79 ± 0.64 | 8.73 ± 4.85 | <0.001*** | 0.018* |
| TC (mmol/L) | 5.14 ± 0.93 | 4.99 ± 1.19 | 0.068 | 1 |
| TG (mmol/L) | 1.50 ± 1.12 | 1.71 ± 1.22 | <0.001*** | 0.018* |
| HDL-C (mmol/L) | 1.49 ± 0.37 | 1.11 ± 0.37 | <0.001*** | 0.018* |
| LDL-C (mmol/L) | 2.83 ± 0.78 | 2.85 ± 0.97 | 0.764 | 1 |
| APOA1 (g/L) | 1.29 ± 0.19 | 1.13 ± 0.28 | <0.001*** | 0.018* |
| APOB (g/L) | 0.88 ± 0.20 | 0.89 ± 0.25 | 0.396 | 1 |

**Notes.**

TP, total protein; ALB, albumin; GLB, globulin; ALB/GLB, albumin/globulin; BUN, blood urea nitrogen; Scr, serum creatinine; SUA, serum uric acid; FBG, fasting blood glucose; TC, total cholesterol; TG, triglyceride; HDL-C, high density lipoprotein cholesterol; LDL-C, low density lipoprotein cholesterol; APOA1, apolipoprotein A1; APOB, apolipoprotein B; P'-value, adjusted for age and gender.

*$P < 0.05$
**$P < 0.01$
***$P < 0.001$

A1298C polymorphisms were significantly associated with the development of T2DM. The frequencies of the TT genotype ($P = 0.030$) of MTHFR C677T and the CC genotype ($P = 0.032$) of MTHFR A1298C were significantly higher in patients with T2DM than in the control group. In addition, patients with T2DM presented a significantly higher frequency of the T allele ($P = 0.047$) of MTHFR C677T. The risk assessments also found that homozygous mutants of MTHFR C677T and A1298C had a 1.750 and 3.132 increased odds of developing T2DM, respectively. Meanwhile, both MTHFR C677T and A1298C polymorphisms were found to be related to T2DM in the recessive model, which conferred 1.608- and 2.988-fold odds, respectively. In addition, logistic regression model showed that MTHFR C677T locus TT genotype and A1298C locus CC genotype could increase the odds of T2DM (Table S2). These results indicated that homozygous mutants of MTHFR C677T (TT genotype) and MTHFR A1298C (CC genotype) appeared to be potential risk factors for developing T2DM in the Dali area of Yunnan Province.

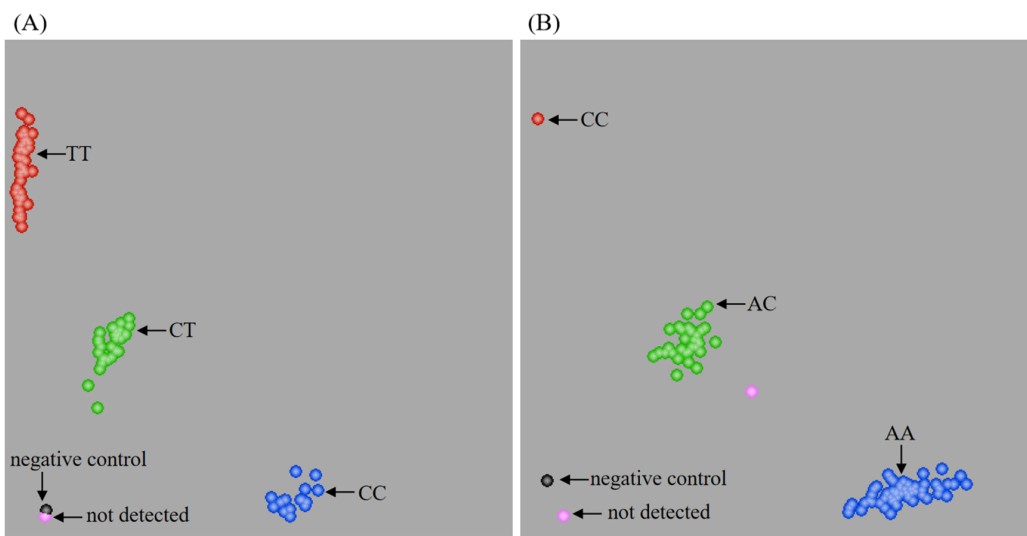

**Figure 1** Genotype representation of MTHFR C677T (A) and A1298C (B) loci by KASP.

**Table 2 Genotype and allele distribution of MTHFR C677T and A1298C polymorphisms in the study population and risk analyses of developing T2DM.**

| Genotypes/Alleles | Control ($n = 272$) $n$ (%) | T2DM ($n = 445$) $n$ (%) | OR (95% CI) | $P$-value |
|---|---|---|---|---|
| MTHFR C677T | | | | |
| CC | 110 (40.44) | 156 (35.06) | 1.000 (reference) | – |
| CT | 135 (49.63) | 222 (49.89) | 1.160 (0.838–1.604) | 0.371 |
| TT | 27 (9.93) | 67 (15.06) | 1.750 (1.052–2.911) | 0.030[*] |
| C | 355 (65.26) | 534 (60.00) | 1.000 (reference) | – |
| T | 189 (34.74) | 356 (40.00) | 1.252 (1.003–1.563) | 0.047[*] |
| Dominant model | – | – | 1.258 (0.922–1.716) | 0.148 |
| Recessive model | – | – | 1.608 (1.000–2.586) | 0.048[*] |
| Overdominant model | – | – | 0.990 (0.732–1.338) | 0.947 |
| MTHFR A1298C | | | | |
| AA | 180 (66.18) | 273 (61.35) | 1.000 (reference) | – |
| AC | 88 (32.35) | 153 (34.38) | 1.146 (0.830–1.583) | 0.407 |
| CC | 4 (1.47) | 19 (4.27) | 3.132 (1.048–9.357) | 0.032[*] |
| A | 448 (82.35) | 699 (78.54) | 1.000 (reference) | – |
| C | 96 (17.65) | 191 (21.46) | 1.275 (0.971–1.674) | 0.080 |
| Dominant model | – | – | 1.233 (0.899–1.690) | 0.193 |
| Recessive model | – | – | 2.988 (1.006–8.879) | 0.039[*] |
| Overdominant model | – | – | 0.913 (0.662–1.258) | 0.577 |

Notes.
OR, adds ratios; 95% CI, 95% confidence interval.
[*]$P < 0.05$

## Association of different genotype combinations of MTHFR C677T and A1298C with the risk of developing T2DM

The MTHFR C677T and A1298C polymorphisms can form nine different genotype combinations: CC/AA, CC/AC, CC/CC, CT/AA, CT/AC, CT/CC, TT/AA, TT/AC, and

**Table 3** Distribution of the different genotype combinations in the study population.

| Genotypes | Control (n = 272) n (%) | T2DM (n = 445) n (%) | OR (95% CI) | P-value |
|---|---|---|---|---|
| CC/AA | 64 (23.53) | 77 (17.30) | 1.000 (reference) | – |
| CC/AC | 44 (16.18) | 68 (15.28) | 1.285 (0.776–2.126) | 0.330 |
| CC/CC | 2 (0.74) | 11 (2.47) | 4.571 (0.977–21.381) | 0.036[*] |
| CT/AA | 89 (32.72) | 140 (31.46) | 1.307 (0.855–2.000) | 0.216 |
| CT/AC | 44 (16.18) | 77 (17.30) | 1.455 (0.885–2.391) | 0.139 |
| CT/CC | 2 (0.74) | 5 (1.12) | 2.078 (0.390–11.071) | 0.382 |
| TT/AA | 27 (9.93) | 56 (12.58) | 1.724 (0.978–3.037) | 0.058 |
| TT/AC | 0 (0.00) | 8 (1.80) | 1.104 (1.031–1.182) | 0.012[*] |
| TT/CC | 0 (0.00) | 3 (0.67) | 1.039 (0.995–1.085) | 0.117 |

Notes.

OR, adds ratios; 95% CI, 95% confidence interval.

[*] $P < 0.05$

TT/CC. The results of the interaction between variants of the MTHFR gene demonstrated that the CC/CC ( $P = 0.036$) and TT/AC ($P = 0.012$) combinations significantly increased the odds of developing T2DM by 4.571- and 1.104-fold, respectively (Table 3).

## Linkage disequilibrium and haplotype analysis of the MTHFR C677T and A1298C polymorphisms

As shown in Fig. 2, there existed linkage disequilibrium between the MTHFR C677T and A1298C polymorphisms in the study population. Haplotype analysis was performed for the MTHFR gene polymorphisms, and four different haplotypes, C-A, T-A, C-C, and T-C, could be formed between the MTHFR C677T and A1298C polymorphisms. The C-A, T-A, and C-C haplotypes were used for analysis because the frequency of the T-C haplotype was less than 0.03. Compared with the control group, the distribution frequency of the T-A haplotype was significantly higher in the T2DM group and conferred a 1.305-fold increased odds of developing T2DM ($P = 0.030$) (Table 4).

## Association of the MTHFR C677T and A1298C polymorphisms with biochemical variables in patients with T2DM

It was found that there were significant differences in the levels of FBG, TG, HDL-C, and APOA1 between the control and T2DM groups. Next, we analyzed the effects of the MTHFR C677T and A1298C polymorphisms on the glycolipid metabolism index alterations in the T2DM group. As shown in Table 5, the patients with T2DM and the CT/TT genotypes of MTHFR C677T or the CC genotype of MTHFR A1298C presented with significantly higher FBG levels, but no differences were found in the lipid levels (Table 5).

## Association of MTHFR C677T and A1298C polymorphisms with Hcy level, inflammatory response, and oxidative stress in patients with T2DM

We next evaluated the Hcy level, inflammatory response, and oxidative stress in patients with T2DM and analyzed their relationships with the MTHFR C677T and A1298C polymorphisms. As expected, the MTHFR C677T or A1298C polymorphisms were significantly associated with increased Hcy levels in patients with T2DM. Moreover, it

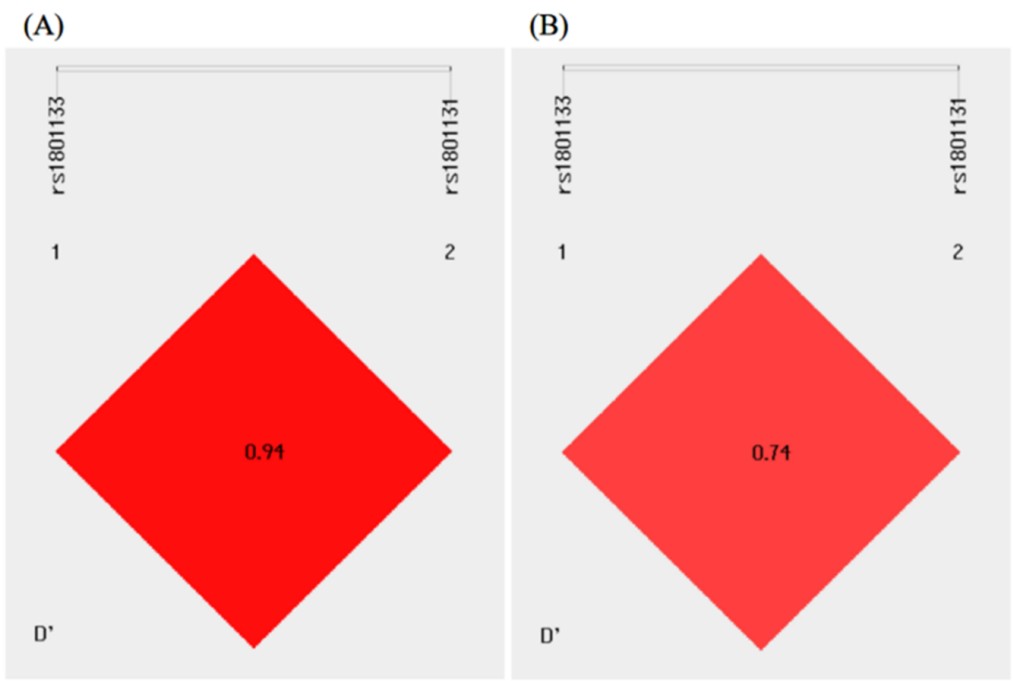

**Figure 2  Linkage disequilibrium analysis of the MTHFR C677T and A1298C polymorphisms in the control (A) and T2DM (B) groups.**

**Table 4  Haplotype analysis of the MTHFR C677T and A1298C polymorphisms in the study population.**

| Haplotypes | Control ($n = 544$) n (%) | T2DM ($n = 890$) n (%) | OR (95% CI) | P-value |
|---|---|---|---|---|
| C-A | 260 (47.79) | 363 (40.79) | 1.000 (reference) | – |
| T-A | 185 (34.01) | 337 (37.87) | 1.305 (1.027–1.658) | 0.030[*] |
| C-C | 97 (17.83) | 171 (19.21) | 1.263 (0.939–1.697) | 0.122 |

Notes.
OR, adds ratios; 95% CI, 95% confidence interval.
[*]$P < 0.05$

was found that the CT and TT genotypes of MTHFR C677T significantly elevated the level of TNF-$\alpha$ (Table 6). Previous studies have demonstrated that Hcy level could be affected by several medications, including metformin, and fibric acid derivatives (*De Jager et al., 2010*; *Herrmann et al., 2012*; *Ntaios et al., 2011*). Our results also showed that the Hcy level increased in the taking metformin or fibric acid derivatives groups, but the differences were not statistically significant ($P > 0.05$) (Tables S3 and S4).

## Association of the MTHFR C677T and A1298C polymorphisms with cardiovascular disease in the T2DM group

We further analyzed the distribution of genotype and allele frequencies of the MTHFR C677T and A1298C loci in patients with T2DM and T2DM with CVD. The results revealed that the frequencies of the CT and TT genotypes and the T allele of MTHFR C677T were

**Table 5** Comparison of the FBG and blood lipid levels of different genotypes in the T2DM group.

| Variables | MTHFR C677T | | | MTHFR A1298C | | |
|---|---|---|---|---|---|---|
| | CC ($n = 156$) | CT ($n = 222$) | TT ($n = 67$) | AA ($n = 273$) | AC ($n = 153$) | CC ($n = 19$) |
| FBG (mmol/L) | 8.99 ± 3.80 | 10.00 ± 4.35 | 10.57 ± 5.15 | 9.59 ± 4.25 | 9.68 ± 4.14 | 12.48 ± 5.84 |
| P-value | 1 | 0.020** | 0.026** | 1 | 0.825 | 0.047* |
| TC (mmol/L) | 5.00 ± 1.10 | 5.04 ± 1.25 | 4.83 ± 1.16 | 4.99 ± 1.25 | 5.03 ± 1.09 | 4.74 ± 0.96 |
| P-value | 1 | 0.730 | 0.305 | 1 | 0.754 | 0.397 |
| TG (mmol/L) | 2.24 ± 2.03 | 2.31 ± 2.06 | 2.03 ± 1.10 | 2.23 ± 1.97 | 2.31 ± 1.97 | 1.86 ± 0.85 |
| P-value | 1 | 0.741 | 0.430 | 1 | 0.684 | 0.422 |
| HDL-C (mmol/L) | 1.17 ± 0.36 | 1.14 ± 0.35 | 1.16 ± 0.32 | 1.15 ± 0.33 | 1.15 ± 0.37 | 1.22 ± 0.46 |
| P-value | 1 | 0.910 | 0.205 | 1 | 0.876 | 0.373 |
| LDL-C (mmol/L) | 2.85 ± 0.89 | 2.89 ± 1.02 | 2.73 ± 0.99 | 2.84 ± 1.01 | 2.90 ± 0.89 | 2.51 ± 1.02 |
| P-value | 1 | 0.690 | 0.396 | 1 | 0.549 | 0.168 |
| APOA1 (g/L) | 1.16 ± 0.24 | 1.13 ± 0.24 | 1.13 ± 0.24 | 1.14 ± 0.24 | 1.14 ± 0.23 | 1.17 ± 0.29 |
| P-value | 1 | 0.183 | 0.471 | 1 | 0.946 | 0.572 |
| APOB (g/L) | 0.88 ± 0.22 | 0.91 ± 0.26 | 0.87 ± 0.24 | 0.89 ± 0.26 | 0.90 ± 0.23 | 0.86 ± 0.22 |
| P-value | 1 | 0.284 | 0.856 | 1 | 0.437 | 0.652 |

Notes.
OR, adds ratios; 95% CI, 95% confidence interval.
*$P < 0.05$
**$P < 0.01$

**Table 6** Comparison of the levels of Hcy, TNF-$\alpha$, IL-6, SOD, and MDA of the different genotypes in the T2DM group.

| Variables | MTHFR C677T | | | MTHFR A1298C | | |
|---|---|---|---|---|---|---|
| | CC ($n = 60$) | CT ($n = 104$) | TT ($n = 42$) | AA ($n = 136$) | AC ($n = 66$) | CC ($n = 4$) |
| Hcy (nmol/mL) | 6.61 ± 1.68 | 8.37 ± 3.47 | 9.81 ± 3.91 | 7.55 ± 2.89 | 9.37 ± 3.90 | 8.45 ± 3.04 |
| P-value | 1 | <0.001*** | <0.001*** | 1 | 0.001** | 0.544 |
| TNF-$\alpha$ (pg/mL) | 59.29 ± 21.79 | 73.81 ± 27.15 | 71.88 ± 22.01 | 66.35 ± 16.47 | 66.96 ± 26.21 | 88.46 ± 17.71 |
| P-value | 1 | 0.002** | 0.037* | 1 | 0.300 | 0.185 |
| IL-6 (pg/mL) | 55.31 ± 13.98 | 52.34 ± 19.60 | 50.22 ± 13.10 | 52.34 ± 12.72 | 56.15 ± 31.21 | 52.98 ± 2.97 |
| P-value | 1 | 0.179 | 0.082 | 1 | 0.469 | 0.764 |
| SOD (U/mL) | 56.90 ± 22.90 | 74.44 ± 25.94 | 67.14 ± 22.81 | 70.32 ± 24.66 | 64.72 ± 25.04 | 54.64 ± 17.34 |
| P-value | 1 | 0.059 | 0.268 | 1 | 0.452 | 0.152 |
| MDA ($\mu$mol/L) | 3.60 ± 3.22 | 4.18 ± 2.69 | 4.55 ± 2.97 | 4.24 ± 3.09 | 3.78 ± 2.61 | 4.09 ± 0.75 |
| P-value | 1 | 0.219 | 0.133 | 1 | 0.303 | 0.924 |

Notes.
FBG, fasting blood glucose; FMN, fructosamine; TC, total cholesterol; TG, triglyceride; HDL-C, high density lipoprotein-cholesterol; LDL-C, low density lipoprotein-cholesterol; APOA1, apolipoprotein A1; APOB, apolipoprotein B; Hcy, homocysteine; TNF-$\alpha$, tumor necrosis factor alpha; IL-6, interleukin 6; SOD, superoxide dismutase; MDA, malondialdehyde.
*$P < 0.05$
**$P < 0.01$
***$P < 0.001$

higher in the T2DM with cardiovascular disease (CVD) group, but the differences were not significant (Table 7). In addition, logistic regression model showed that MTHFR C677T and A1298C polymorphisms had no significant effect in patients with T2DM and CVD (Table S5). Next, we analyzed the genotype and allele frequencies of the MTHFR C677T

**Table 7  Genotype and allele frequency distribution of the MTHFR C677T and A1298C polymorphisms in the T2DM and T2DM with CVD groups.**

| Genotypes/Alleles | T2DM ($n = 198$) $n$ (%) | T2DM with CVD ($n = 247$) $n$ (%) | OR (95% CI) | $P$-value |
|---|---|---|---|---|
| MTHFR C677T | | | | |
| CC | 75 (37.88) | 82 (33.20) | 1.000 (reference) | – |
| CT | 94 (47.47) | 126 (51.01) | 1.226 (0.812–1.850) | 0.332 |
| TT | 29 (14.65) | 39 (15.79) | 1.230 (0.693–2.183) | 0.479 |
| C | 244 (61.62) | 290 (58.70) | 1.000 (reference) | – |
| T | 152 (38.38) | 204 (41.30) | 1.129 (0.862–1.480) | 0.378 |
| MTHFR A1298C | | | | |
| AA | 118 (59.60) | 156 (63.16) | 1.000 (reference) | – |
| AC | 71 (35.86) | 80 (32.39) | 0.852 (0.572–1.270) | 0.432 |
| CC | 9 (4.54) | 11 (4.45) | 0.925 (0.371–2.303) | 0.866 |
| A | 307 (77.53) | 392 (79.35) | 1.000 (reference) | – |
| C | 89 (22.47) | 102 (20.65) | 0.898 (0.651–1.237) | 0.509 |

Notes.
OR,  adds ratios;  95% CI,  95% confidence interval.

and A1298C polymorphisms in patients with T2DM and T2DM with CVD with diabetes for 0–5 years and >5 years. In patients diagnosed with diabetes for 0–5 years, the CT genotype frequency of the MTHFR C677T locus was higher in patients with T2DM and CVD. In patients with diabetes for >5 years, the frequencies of the CT and TT genotypes and the T allele of the MTHFR C677T locus, and the frequency of the CC genotype of the MTHFR A1298C locus were higher in patients with T2DM and CVD, but there were no significant differences between these two groups (Table 8).

## DISCUSSION

Numerous studies have been conducted considering the association of genetic polymorphisms with the development of T2DM in various populations, which have revealed controversial results. To the best of our knowledge, this is the first study to evaluate the association of the MTHFR C677T (rs1801133) and A1298C (rs1801131) polymorphisms with the risk of developing T2DM in the population of the Dali area of Yunnan Province, China. We identified that the MTHFR C677T and A1298C polymorphisms significantly increase the risk of developing T2DM among the population in the Dali area.

There are various studies describing the association between MTHFR C677T and T2DM susceptibility. *Sarkar, Chatterjee & Bandyopadhyay (2021)* collected samples from 104 patients with T2DM and 176 healthy controls and found that the CT genotype of MTHFR C677T exhibited a significantly higher risk of developing T2DM compared to the CC genotype in the Bengalee Hindu caste population of West Bengal, India. A meta-analysis on 6,783 subjects published by *Zhu et al. (2014)* showed that the MTHFR C677T locus T allele may be a genetic risk factor of T2DM in the Chinese population. There was the same association between MTHFR C677T polymorphism and T2DM susceptibility in the Middle East and North Africa region population through a meta-analysis (*El Alami et*

Liu et al. (2024), *PeerJ*, DOI 10.7717/peerj.18334

**Table 8   Genotype and allele frequency distribution of the MTHFR C677T and A1298C polymorphisms in patients with T2DM and T2DM with CVD who have been diagnosed with diabetes for 0–5 years and >5 years.**

| Genotypes /Alleles | 0–5 years | | OR (95% CI) | *P*-value | >5 years | | OR (95% CI) | *P*-value |
|---|---|---|---|---|---|---|---|---|
| | T2DM ($n = 106$) n (%) | T2DM with CVD ($n = 115$) n (%) | | | T2DM ($n = 90$) n (%) | T2DM with CVD ($n = 134$) n (%) | | |
| MTHFR C677T | | | | | | | | |
| CC | 37 (34.91) | 39 (33.91) | 1.000 (reference) | – | 36 (40.00) | 44 (32.84) | 1.000 (reference) | – |
| CT | 56 (52.83) | 63 (54.78) | 1.067 (0.600–1.899) | 0.825 | 38 (42.22) | 64 (47.76) | 1.378 (0.759–2.501) | 0.291 |
| TT | 13 (12.26) | 13 (11.31) | 0.949 (0.389–2.312) | 0.908 | 16 (17.78) | 26 (19.40) | 1.330 (0.620–2.851) | 0.464 |
| C | 130 (61.32) | 141 (61.30) | 1.000 (reference) | – | 110 (61.11) | 152 (56.72) | 1.000 (reference) | – |
| T | 82 (38.68) | 89 (38.70) | 1.001 (0.682–1.468) | 0.997 | 70 (38.89) | 116 (43.28) | 1.199 (0.816–1.762) | 0.355 |
| MTHFR A1298C | | | | | | | | |
| AA | 61 (57.55) | 67 (58.26) | 1.000 (reference) | – | 56 (62.22) | 91 (67.91) | 1.000 (reference) | – |
| AC | 40 (37.73) | 43 (37.39) | 0.979 (0.563–1.701) | 0.939 | 31 (34.45) | 38 (28.36) | 0.754 (0.423–1.347) | 0.340 |
| CC | 5 (4.72) | 5 (4.35) | 0.910 (0.251–3.298) | 0.886 | 3 (3.33) | 5 (3.73) | 1.026 (0.236–4.459) | 0.973 |
| A | 162 (76.42) | 177 (76.96) | 1.000 (reference) | – | 143 (79.44) | 220 (82.09) | 1.000 (reference) | – |
| C | 50 (23.58) | 53 (23.04) | 0.970 (0.624–1.508) | 0.893 | 37 (20.56) | 48 (17.91) | 0.843 (0.523–1.360) | 0.484 |

**Notes.**
OR, adds ratios; 95% CI, 95% confidence interval.

*al., 2024*). However, *El Hajj Chehadeh et al. (2016)* collected 169 T2DM patients and 209 healthy controls and their findings demonstrate that the MTHFR gene polymorphisms are not related to T2DM in the Emirati population. In addition, no association was also observed between MTHFR polymorphism and T2DM in 118 Taiwanese or 359 Bahraini (*Al-Harbi et al., 2015*; *Chang et al., 2011*). Polymorphisms of the MTHFR gene vary widely by area and ethnicity, thus suggesting that any association between SNPs and T2DM risk may likewise depend on the area and population ethnicity. This case-control study revealed an obvious association of the MTHFR C677T and A1298C polymorphisms with T2DM susceptibility in the population of the Dali area, which was similar to the findings of *Zhu et al. (2014)*. To investigate the interactive effects of the derived allele combinations on T2DM risk, haplotype analysis was performed for the MTHFR C677T and A1298C loci. We found that the T-A haplotype conferred an increased risk of developing T2DM in the population of the Dali area. Our results provide strong evidence that MTHFR C677T and A1298C are significantly associated with the development of T2DM in the population of the Dali area. Thus, it is necessary to clarify the roles of these SNPs in different areas.

T2DM is a complex, heterogeneous, and polygenic metabolic disease that is characterized by hyperglycemia and is accompanied by disorders of fat and protein metabolism (*Abbas et al., 2013*). Studies have found that dyslipidemia increases the risk of developing T2DM (*Fang et al., 2022*; *Lee et al., 2022*; *Gheflati et al., 2019*). Accordingly, significant changes were observed in the levels of FBG, TG, HDL-C, and APOA1 between the T2DM and control groups. It has also been reported that MTHFR C677T could affect LDL-C levels (*Yilmaz et al., 2003*; *Pitsavos et al., 2006*), and our findings indicate an association of MTHFR C677T and A1298C polymorphisms with FBG levels in patients with T2DM in the Dali area, Yunnan Province. However, no significant differences were found between MTHFR gene polymorphisms and the lipid metabolism parameters.

Individuals with T2DM often have a persistent low-grade inflammatory response and oxidative stress. Previous research has suggested that proinflammatory cytokines, such as TNF-$\alpha$ and interleukins (ILs), could impair insulin signaling and cause insulin resistance (*Hotamisligil, Shargill & Spiegelman, 1993*; *Krogh-Madsen et al., 2006*; *Fain, 2006*). The reactive oxygen species produced by oxidative stress, such as hydroxyl radicals, superoxide anions, and peroxides, have been linked to the development of insulin resistance and $\beta$-cell dysfunction (*Babel & Dandekar, 2021*; *Rains & Jain, 2011*). These inflammatory mediators and free radicals could also cause direct damage to vascular endothelial cells and accelerate the development of vascular diseases. In this study, we identified that the MTHFR C677T polymorphism was significantly associated with elevated TNF-$\alpha$ levels in patients with T2DM. Although no significant association was observed in the variables of oxidative stress, the increased level of MDA implied the potential role of MTHFR C677T in oxidative stress.

CVD is a common complication of diabetes and is a major cause of death and disability among individuals with diabetes (*Napoli et al., 2020*; *Einarson et al., 2018*). Studies have shown that the risk of CVD in patients with T2DM was 2–4 times higher than that in individuals without T2DM (*Tuttle et al., 2021*). Thus, a major goal in the prevention and control of T2DM is to reduce the risk of cardiovascular events. Early identification or

screening of populations at high risk of developing T2DM and T2DM-related cardiovascular complications might provide a strategy for the early prevention of diabetes. *Settin et al. (2015)* found that the TT genotype of the MTHFR C677T locus could increase the risk of developing T2DM, and the CT and TT genotypes could increase the risk of developing diabetes complications such as peripheral neuropathy and ischemic heart disease. Furthermore, Hcy is a widely used biomarker for predicting cardiovascular events in patients with T2DM (*Guieu, Ruf & Mottola, 2022*; *Raghubeer & Matsha, 2021*). We found that patients with T2DM, who also had the high diabetic risk genotypes of MTHFR C677T and A1298C, had significantly higher Hcy levels, which suggests a potential increased risk of cardiovascular events. A sustained high level of Hcy could induce vascular endothelial injury, promote oxidative stress, and form atherosclerotic plaques, which may eventually result in cardiovascular events (*Gospodarczyk et al., 2022*). Furthermore, we analyzed the association of MTHFR gene polymorphisms with the risk of CVD in patients with T2DM. The results revealed that the frequencies of the CT and TT genotypes and the T allele of MTHFR C677T were higher in the patients with T2DM and CVD group compared to the patients with T2DM group, but no significant difference was found in these two groups. However, a further large-sample prospective study is required to verify these results.

In conclusion, we demonstrated that the MTHFR C677T and A1298C polymorphisms are important determinants for the incidence of T2DM in the population of the Dali area of Yunnan Province, China. Our findings provide additional information for the prevention and treatment of T2DM through folate supplementation in the local region. The sample size of this study was small, which may have a certain impact on the statistical results. In the future, we will continue to expand the sample size to explore the relationship between MTHFR C677T and A1298C polymorphism and the development of T2DM in Dali area of Yunnan Province, which has certain referencevalue for the early screening of high-risk groups in this area, and provides a new ideafor the prevention of T2DM patients.

## ACKNOWLEDGEMENTS

We are thankful to all the study participants for their contribution and cooperation toward thiswork. We thank the reviewers for their insightful comments on this article.

### Funding

This work was financially supported by the Undergraduate Research Fund of Dali University in 2022 (Grant No. 84) and the National Undergraduate Innovation and Entrepreneurship Training Program (Grant No. 202110679036). The funders had no role in study design, data collection and analysis, decision to publish, or preparation of the manuscript.

### Grant Disclosures

The following grant information was disclosed by the authors:
The Undergraduate Research Fund of Dali University: 84.

National Undergraduate Innovation and Entrepreneurship Training Program: 202110679036.

## Competing Interests
The authors declare there are no competing interests.

## Author Contributions
- Yongxin Liu performed the experiments, analyzed the data, prepared figures and/or tables, and approved the final draft.
- Genyuan Pu performed the experiments, analyzed the data, prepared figures and/or tables, and approved the final draft.
- Caiting Yang performed the experiments, prepared figures and/or tables, and approved the final draft.
- Yuqing Wang performed the experiments, prepared figures and/or tables, and approved the final draft.
- Kaitai Jin performed the experiments, prepared figures and/or tables, and approved the final draft.
- Shengrong Wang performed the experiments, prepared figures and/or tables, and approved the final draft.
- Xiao Liang performed the experiments, prepared figures and/or tables, and approved the final draft.
- Shenghe Hu performed the experiments, prepared figures and/or tables, and approved the final draft.
- Shuguang Sun conceived and designed the experiments, authored or reviewed drafts of the article, and approved the final draft.
- Mingming Lai conceived and designed the experiments, authored or reviewed drafts of the article, and approved the final draft.

## Human Ethics
The following information was supplied relating to ethical approvals (*i.e.*, approving body and any reference numbers):

This study was approved by the Medical Ethics Committee of Dali University (Approval No. 202010-21).

## Data Availability
The raw data are available in the Supplemental Files.

## Supplemental Information
Supplemental information for this article can be found online at http://dx.doi.org/10.7717/peerj.18334#supplemental-information.

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
