# Peer review of "Association analysis of MTHFR (rs1801133 and rs1801131) gene polymorphism towards the development of type 2 diabetes mellitus in Dali area population from Yunnan Province, China"

_PeerJ, doi:10.7717/peerj.18334_

## Round 0.1 · original submission · Major Revisions

Please, edit the manuscript according remarks of the Reviewers. Pay special attention to statistical analysis.

Reviewer 1 ·

Basic reporting

The MS “Association analysis of MTHFR (rs1801133 and rs1801131)gene polymorphism towards the development of type 2diabetes mellitus in Dali area population from YunnanProvince, China” is a research paper with an appropriate design. The aim of the study this study was to investigate the effects of MTHFR C677T (rs1801133) and A1298C (rs1801131) loci
polymorphisms on T2DM susceptibility in Dali area population of Yunnan Province, China. 
Their study suggested that genetic variations of MTHFR C677T and A1298C were significantly associated with susceptibility to T2DM in Dali area population from Yunnan Province, China..
Major problem of the study is the sample size which is questionable with regard to the genotype distribution (very low percentage of risk genotypes, especially MTHF A1298C). Moreover, haplotype analysis may also be problematic with these relatively low numbers of risk alleles-genotypes. All the results are derived from the data of rather low sample size
Comment 1: Abstract is informative, rather well written, however the Results part of the Abstract section is without any data, numbers.
Comment 2: In the Abstract section authors used too many abbreviations.
Comment 3: Introduction section is generally well written.
Comment 4: Discussion is not optimally written. Authors mentioned several studies in various populations, and they did not use references. Moreover, they should compare their study with other studies (numbers of cases and controls...)

Comment 5: Language shoud be improved.

Experimental design

Major problem of the study is the sample size which is questionable with regard to the genotype distribution (very low percentage of risk genotypes, especially MTHF A1298C). Moreover, haplotype analysis may also be problematic with these relatively low numbers of risk alleles-genotypes. All the results are derived from the data of rather low sample size

Validity of the findings

Major problem of the study is the sample size which is questionable with regard to the genotype distribution (very low percentage of risk genotypes, especially MTHF A1298C). Moreover, haplotype analysis may also be problematic with these relatively low numbers of risk alleles-genotypes. All the results are derived from the data of rather low sample size

Reviewer 2 ·

Basic reporting

This paper describes an association study for 2 previously described variants (rs1801133 and rs180131) in the MTHFR gene in 445 individuals with type 2 diabetes and 272 controls from the Dali area from Yunnan Province China. The introduction should give adjusted p values and odds ratios rather than use words such as “considerably associated.”

Experimental design

The experimental design is flawed in that there is no power calculation to support that this incredibly small sample has enough power to detect the effect of a SNP for the various traits.

Many independent traits are analyzed yet no Bonferroni correction is provided.

No adjustments (e.g. age, sex, PCs) are given.

BMI is not provided for subjects.

No information is given on treatment of individuals with type 2 diabetes which could confound associations such as fasting blood glucose.

Justification is not given for many of the traits which were analyzed.

Validity of the findings

Overall, the findings are not convincing due to the borderline, unadjusted p values and small numbers of subjects. The authors must recognize that odds ratios become inflated in very small samples.

It is concerning that some of the associations were only significant when analyzed as a recessive which further reduced the sample size. This was also true of the haplotype analysis.

Independent replication in another sample would have made these results more credible.

Additional comments

In general, I commend the authors for wanting to contribute to our understanding of the genetic basis for type 2 diabetes across various ethnic groups. However, what we’ve learned over the past 20 years is that very large datasets are needed, and associations detected in extremely underpowered samples typically become non-significant when the sample size is increased.

Reviewer 3 ·

Basic reporting

a. Overall, the manuscript is well-organized and the introduction provides context, background information, and aims of the study.
b. The aspect ratio of the figures is elongated, which seem to distort the appearance of data. Adjustment of dimensions is needed to maintain the aspect ratio and visual clarity. (Ensure that figures are scaled appropriately without stretching or compressing the data).
c. The tables lack footnotes that explain abbreviations, and statistical analysis.
d. The provided raw data file only includes genotypes without clear labeling. Other underlying data are not provided.
e. There are areas where the language could be refined to enhance clarity and precision. For instance, the introduction contains sentences that could be rephrased for greater clarity of the main arguments. I suggest reviewing the introduction (as well as the discussion) ensuring that each sentence clearly communicates its intended meaning using grammatically correct scientific professional language (professional language editing is needed).

Experimental design

This original research article falls within the Journal aims and scope, research question is well defined and stated.
However, several points and details need to be added to the methodology section and clarified to ensure the replicability and validity of the results and conclusion:
1. Inclusion and exclusion criteria for the healthy controls.
2. Diagnosis criteria for cardiovascular disease in T2DM patients.
3. Power analysis and sample size calculation.
4. Medication history for T2DM patients.
5. more details are included in additional comments section.

Validity of the findings

* The raw data provided only includes genotypes without clear labeling. Other underlying data are not provided.
* An inherent limitation of case-control studies is the lack of temporality (i.e., they cannot establish the sequence of events (disease and risk factors)). Therefore case-control studies do not assess the risk of disease, but rather the odds of it. Additionally, case-control studies do not predict incidence nor susceptibility (incidence parameter requires prospective study design). Accordingly, the conclusions drawn from this study should be modified in both the main text (e.g. line 270) and the abstract.

Additional comments

1. Consider adding context and literature review regarding the association between studied polymorphism and oxidative stress in the introduction.
2. Please provide the rationale behind the determination of the sample size for both the T2DM patient group and the healthy control group. Which statistical parameters (power analysis) were used in the sample size calculation (e.g., effect size, variability, power, significance level)? Please specify the values used or refer to the equations/methods/references employed.
3. Please specify the criteria used to identify and select healthy controls and describe their recruitment process. This could include age range, gender distribution, and any relevant demographic factors. Was there any matching criteria used in relation to the T2DM patient group?
4. Homocysteine levels are affected by several medications, including metformin, and dyslipidemia drugs (fibric acid derivatives). Provide the drug history of patients (>50% have hyperlipidemia) and/or controls, in the result section (or methods if these were among inclusion/exclusion criteria).
5. In the methods section, lines #105-109, provide the units of measurement for each biochemical parameter (e.g., mmol/L for glucose, mg/dL for cholesterol) and introduce the term then the abbreviation in brackets to maintain consistency and clarity throughout the manuscript (e.g. total protein (TP)).
6. The results include analysis of patients with CVD, Please provide criteria and methods used to diagnose CVD.
7. I would like to suggest considering the inclusion of measurements for B12 and folate levels, along with an analysis of their correlation with specific genotypes (C677T, A1298C).
8. These type of results are subject to bias and confounding, Using a regression model to adjust for potential confounders could add evidence and strengthen the conclusions. Consider adding a regression model to predict the contributing role of the studied genotypes in the odds of T2DM (and CVD complications) with adjustment of potential confounders.
9. Please consider adding footnotes for tables to enhance clarity and provide additional context for readers. This will help ensure that readers have a complete understanding of the information presented in the tables. Footnotes usually explain abbreviations and clarify methods or statistical tests used.
10. describe the limitations and strengths of the study in the discussion section

---

## Round 0.2 · accepted · Accept

The prior reviewers were unavailable to re-review, but I confirm in my opinion that it seems you have followed all recommendations of the reviewers. You have also provided evidence that the study group was sufficiently large. The paper can be accepted.